# Unsplit superconducting and time reversal symmetry breaking transitions in $Sr_2RuO_4$ under hydrostatic pressure and disorder

Vadim Grinenko [1,2✉], Debarchan Das [3], Ritu Gupta[3], Bastian Zinkl[4], Naoki Kikugawa[5], Yoshiteru Maeno [6], Clifford W. Hicks [7,8], Hans-Henning Klauss [1], Manfred Sigrist[4✉] & Rustem Khasanov [3✉]

There is considerable evidence that the superconducting state of $Sr_2RuO_4$ breaks time reversal symmetry. In the experiments showing time reversal symmetry breaking, its onset temperature, $T_{TRSB}$, is generally found to match the critical temperature, $T_c$, within resolution. In combination with evidence for even parity, this result has led to consideration of a $d_{xz} \pm id_{yz}$ order parameter. The degeneracy of the two components of this order parameter is protected by symmetry, yielding $T_{TRSB} = T_c$, but it has a hard-to-explain horizontal line node at $k_z = 0$. Therefore, $s \pm id$ and $d \pm ig$ order parameters are also under consideration. These avoid the horizontal line node, but require tuning to obtain $T_{TRSB} \approx T_c$. To obtain evidence distinguishing these two possible scenarios (of symmetry-protected versus accidental degeneracy), we employ zero-field muon spin rotation/relaxation to study pure $Sr_2RuO_4$ under hydrostatic pressure, and $Sr_{1.98}La_{0.02}RuO_4$ at zero pressure. Both hydrostatic pressure and La substitution alter $T_c$ without lifting the tetragonal lattice symmetry, so if the degeneracy is symmetry-protected, $T_{TRSB}$ should track changes in $T_c$, while if it is accidental, these transition temperatures should generally separate. We observe $T_{TRSB}$ to track $T_c$, supporting the hypothesis of $d_{xz} \pm id_{yz}$ order.

[1] Institute for Solid State and Materials Physics, Technische Universität Dresden, Dresden, Germany. [2] Leibniz-Institut für Festkörper- und Werkstoffforschung (IFW) Dresden, Dresden, Germany. [3] Laboratory for Muon Spin Spectroscopy, Paul Scherrer Institut, Villigen, Switzerland. [4] Institute for Theoretical Physics, ETH Zurich, Zurich, Switzerland. [5] National Institute for Materials Science, Tsukuba, Japan. [6] Department of Physics, Kyoto University, Kyoto, Japan. [7] Max Planck Institute for Chemical Physics of Solids, Dresden, Germany. [8] School of Physics and Astronomy, University of Birmingham, Birmingham, UK. ✉email: vadim.a.grinenko@gmail.com; mansigri@ethz.ch; rustem.khasanov@psi.ch

For unconventional superconductors identifying the symmetry of the order parameter is crucial to pinpoint the origin of the superconductivity. Unconventional pairing states are distinguished from conventional ones by a non-trivial intrinsic phase structure which causes additional spontaneous symmetry breaking at the superconducting phase transition. This can lead, for instance, to a reduction of the crystal symmetry or the loss of time reversal symmetry. Indeed, several superconductors are known, which show experimental responses consistent with time reversal symmetry breaking (TRSB) superconductivity[1–11].

TRSB superconducting states are formed by combining two or more order parameter components with complex coefficients. These components may be degenerate by symmetry, belonging to a single irreducible representation of the crystalline point group (as in the case of $p_x \pm ip_y$ or $d_{xz} \pm id_{yz}$ superconductivity on a tetragonal lattice), or they may come from different representations (for example, $d_{xy} \pm id_{x^2-y^2}$ superconductivity on a tetragonal lattice). In the following, we refer to the former as single-representation and the latter as composite-representation order parameters. For composite-representation order parameters, the two components will generally onset at different temperatures. The higher transition temperature becomes $T_c$, the superconducting critical temperature, and the lower temperature $T_{TRSB}$, the temperature where TRSB onsets. The possibility of composite order parameters is usually dismissed out of hand, because it is unusual for two components that are not related by symmetry to be close enough in energy. However, there are a few known examples: $s$ and $d_{x^2-y^2}$ are relatively close in energy in iron-based superconductors[11,12], while both $(U,Th)Be_{13}$[1,4] and $UPt_3$[2,3,8] have split $T_c$ and $T_{TRSB}$.

Here, we study $Sr_2RuO_4$, an unconventional superconductor[13,14], in which the origin of the superconductivity remains a mystery. Evidence that this superconductor breaks time reversal symmetry comes from zero-field muon spin rotation/relaxation (ZF-$\mu$SR) experiments[15] and polar Kerr effect measurements[16]. Phase-sensitive probes using a corner SQUID device give further support[17]. Moreover, the Josephson effect between a conventional superconductor and $Sr_2RuO_4$ reveal features compatible with the presence of superconducting domains, as expected for TRSB superconductivity[18–20]. For two decades, the leading candidate state to explain these and other observations was the chiral $p$-wave state $p_x \pm ip_y$ (the lattice symmetry of $Sr_2RuO_4$ is tetragonal), which has odd parity and therefore equal spin pairing. However, there is compelling evidence against an order parameter with such spin structure. This evidence includes paramagnetic limiting for in-plane magnetic fields[21–23] and the recently discovered drop in the NMR Knight shift below $T_c$[24,25]. In combination with the above experimental support for TRSB superconductivity, this evidence compels consideration of $d_{xz} \pm id_{yz}$ order.

$d_{xz} \pm id_{yz}$ order would be a surprise because it has a line node at $k_z = 0$, which under conventional understanding requires interlayer pairing, while in $Sr_2RuO_4$ interlayer coupling is very weak. It has been proposed that $d_{xz} \pm id_{yz}$ order might be obtained through multi-orbital degrees of freedom; in this model the order parameter symmetry is encoded in orbital degrees of freedom, so interlayer pairing is not required[26]. This form of pairing is also under consideration for $URu_2Si_2$[27,28]. However, so far it has not been unambiguously confirmed in any material. To avoid horizontal line nodes, the composite-representation order parameters $s \pm id_{x^2-y^2}$[29], $s \pm id_{xy}$[30] and $d_{x^2-y^2} \pm ig_{xy(x^2-y^2)}$[31,32] have also recently been proposed for $Sr_2RuO_4$. In contrast to $d_{xz} \pm id_{yz}$, these require tuning to obtain $T_c \approx T_{TRSB}$ on a tetragonal lattice.

In this work, to test whether the order parameter of $Sr_2RuO_4$ is of single- or composite-representation type we perform ZF-$\mu$SR measurements on hydrostatically pressurised $Sr_2RuO_4$ and on La-doped $Sr_{2-y}La_yRuO_4$. Both of these perturbations maintain the tetragonal symmetry of the lattice. If the order parameter has single-representation nature, $T_{TRSB}$ will track $T_c$. If the order parameter is of the composite-representation kind, with $T_{TRSB}$ matching $T_c$ in clean, unstressed samples through an accidental fine tuning, then perturbations away from this point should in general split $T_{TRSB}$ and $T_c$, whether they preserve tetragonal lattice symmetry or not[33]. Here, we have observed a clear suppression of $T_{TRSB}$ at a rate matching the suppression of $T_c$. Our experimental results provide evidence in favour of single-representation nature of the order parameter in $Sr_2RuO_4$.

## Results

**$\mu$SR on $Sr_2RuO_4$ under hydrostatic pressure.** The hydrostatic pressure measurement setup is shown schematically in Fig. 1. $Sr_2RuO_4$ crystals of diameter $\varnothing \sim 3$ mm were affixed to oxygen-free copper foils, and assembled into an approximately cylindrical collection of total diameter $\varnothing \sim 7$ mm and total length $l$ ~12 mm (see Fig. 1a). The $c$-axes of the separate crystals were aligned to within 3°.

The pressure cell used in the present study (refs. 34,35 and Fig. 1b) is a modification of a "classic $\mu$SR" clamped pressure cell[35,36]. It consists of a main body that encloses the sample and pressure medium, a teflon cap with a metallic support, a tungsten carbide piston, a pressing pad and a clamping bolt (not shown) that holds the piston in place. All the metallic parts of the cell apart from the piston are made from a nonmagnetic beryllium-copper alloy, which is known to have a temperature-independent $\mu$SR response[34–36]. The main feature of this cell is that the only materials placed in the muon beam are the sample, the pressure medium and this CuBe alloy. The muons had a typical momentum of 97 MeV/c, sufficient to penetrate the walls of the pressure cell. The pressure medium was 7373 Daphne oil, which at room temperature solidifies at a pressure $p \approx 2.3$ GPa[37]. The maximum pressure reached here was 0.95 GPa, and therefore hydrostatic conditions are expected. The pressure was determined by monitoring the critical temperature of a small piece of indium (the pressure indicator) placed inside the cell with the $Sr_2RuO_4$ sample. Confirmation that essentially hydrostatic conditions were attained is provided by the fact that $T_c$ was observed to decrease linearly with pressure, whereas in-plane uniaxial stress on a GPa scale causes a strong non-linear increase in $T_c$[38].

The samples used here were grown by the standard floating zone method[39]. Measurements of heat capacity of pieces cut from the ends of the rods used here revealed an average $T_c$ of 1.30(6) K (see Supplementary Fig. 1 in Supplementary Note 1), slightly below the limit of $T_c$ of 1.50 K for a pure sample.

$T_c$ and $T_{TRSB}$ were both obtained by means of $\mu$SR, ensuring that both quantities were measured for precisely the same sample volume. In the $\mu$SR method, spin-polarised muons are implanted, and their spins then precess in the local magnetic field. By collecting statistics of decay positrons in selected direction(s), the muon polarisation as a function of time after implantation, $P_\mu(t)$, can be determined; the time-evolution of this polarisation is determined by the magnetic fields in the sample[40].

$T_c$ is determined through transverse-field (TF) measurements. An external field $B_{ext}$ of 3 mT, as is generated by Helmholtz coils, was applied parallel to the crystalline $c$-axis and perpendicular to the initial muon spin polarisation $\mathbf{P}_\mu(0)$. Measurements were performed in the field-cooled (FC) mode. Details of the method and analysis are given in the "Methods" section.

Example TF-$\mu$SR time spectra at pressure $p = 0.95$ GPa, and at a temperature above $T_c$ and one below, are shown in Fig. 2a. Above $T_c$, the spins of muons stopped in both the sample and the

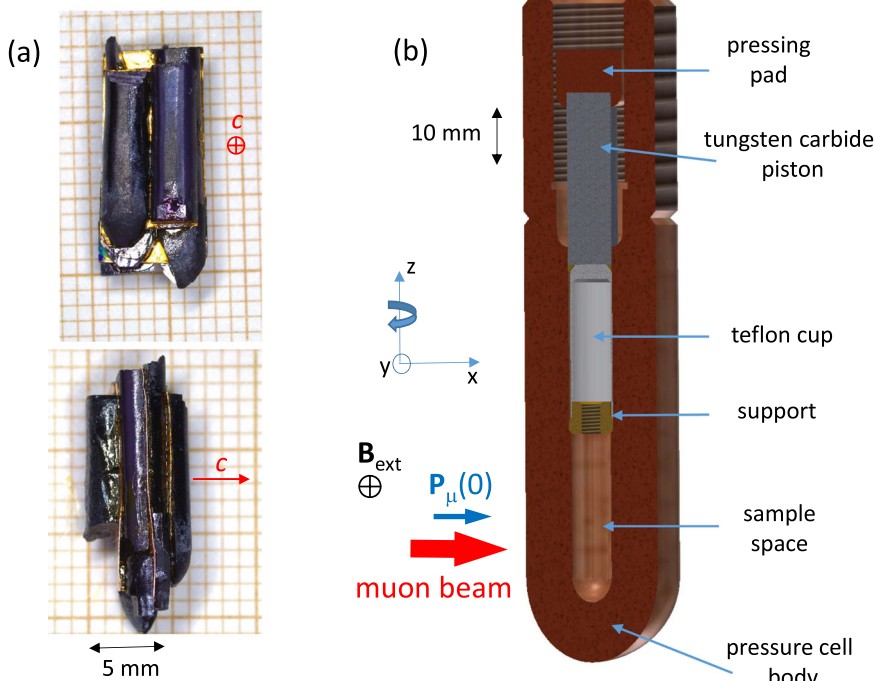

**Fig. 1 Setup for hydrostatic pressure experiments. a** $Sr_2RuO_4$ sample, consisting of semi-cylindrical pieces glued on oxygen-free copper foils. The top and the bottom panels are the front and the side view, respectively. The crossed circle and the arrow indicate the orientation of the $c$-axis. **b** Construction of the pressure cell[34]. The sample and the pressure medium are surrounded only by beryllium-copper (the pressure cell body and the teflon cap support). The parts of the cell with strong $\mu$SR response (teflon cap and tungsten carbide piston) are far from the sample and outside of the muon beam. The initial muon spin polarisation $\mathbf{P}_\mu(0)$ and the external field $\mathbf{B}_{ext}$ in TF-$\mu$SR measurements are aligned along the $x$- and $y$-axes, respectively. By rotating the cell about the $z$-axis, the angle between $\mathbf{P}_\mu(0)$ and the sample $c$-axis can be varied.

pressure cell walls precess with frequency $\omega = \gamma_\mu B_{ext}$ (where $\gamma_\mu = 2\pi \times 135.5$ MHz/T is the muon gyromagnetic ratio). The muon spin polarisation is seen to relax substantially on a 10 $\mu$s time scale. This is because ~50% of muons are implanted into the CuBe, where the nuclear magnetic moments of Cu rapidly relax their polarisation. Below $T_c$, the internal field in the sample becomes highly inhomogeneous due to the appearance of a flux-line lattice, and so the polarisation of the muons that implanted in the sample also relaxes quickly.

TF-$\mu$SR measurements were performed at 0, 0.25, 0.62, and 0.95 GPa. Data at 0 and 0.95 GPa are shown in Fig. 2, and at the other two pressures in Supplementary Figs. 3 and 4 in Supplementary Note 2. Data are analysed as a sum of background and sample contributions, given by Eqs. (3) and (4) (in the "Methods" section), respectively. From the sample contribution we extract a Gaussian relaxation rate, $\sigma$, and the diamagnetic shift of the field inside the sample, $B_{int} - B_{ext} \propto M_{FC}$[41] ($M_{FC}$ is the FC magnetisation). Figure 2b, c, respectively, shows the temperature dependence of $\sigma$ and $B_{int} - B_{ext}$. $\sigma$ is given by $\sigma^2 = \sigma_{sc}^2 + \sigma_{nm}^2$, where $\sigma_{sc}$ and $\sigma_{nm}$ are the flux-line lattice and nuclear moment contributions, respectively. $\sigma_{sc} \propto \lambda_{ab}^{-2}$, where $\lambda_{ab}$ is the in-plane magnetic penetration depth; see ref. [42] and the "Methods" section. The onset of superconductivity can be seen in both $\sigma$ and $B_{int} - B_{ext}$, as a transition rounded on a scale of ~0.1 K. The heat capacity measurements show a similar distribution of $T_c$'s; see Supplementary Fig. 1 in Supplementary Note 1.

The pressure dependence of $T_c$ is shown in Fig. 2g. The error bars in the figure are the rounding on the transition, and can be taken as an absolute error on $T_c$. When fitting $\sigma(T)$ and $B_{int}(T)$ with model functions, the statistical error on the $T_c$'s extracted is considerably smaller, meaning that the error on changes in $T_c$ is low. A linear fit to $T_c(p)$ yields a slope $dT_c/dp = -0.24(2)$ K/GPa,

which is in good agreement with literature data[43–45]. The unpressurised $T_c$ is found to be 1.26(5) K, in good agreement with 1.30(6) K found in the heat capacity measurements, see Supplementary Note 1.

$T_{TRSB}$ is determined through ZF measurements. The signature of TRSB is an enhancement in the muon spin relaxation rate below $T_{TRSB}$, indicating the appearance of spontaneous magnetic fields. In these measurements, external fields were compensated to better than 2 $\mu$T, ruling out flux lines below $T_c$ as the origin of this signal. An example of ZF-$\mu$SR time spectra above and below $T_c$, showing the faster relaxation below $T_c$, at $p = 0.95$ GPa is presented in Fig. 2d. The pressure cell background is $T$-independent, so the increased signal decay comes from the sample. The sample contribution was modelled by a two-component relaxation function: $GKT(t) \cdot \exp(-\lambda t)$, in accordance with the results of refs. [5,6,9,15,46,47]; see also the "Methods" section. Here, $GKT(t)$ is the Gaussian Kubo-Toyabe function describing the relaxation of muon spin polarisation in the random magnetic field distribution created by nuclear magnetic moments, and $\exp(-\lambda t)$ is a Lorentzian decay function accounting for appearance of spontaneous magnetic fields. Temperature dependencies of the exponential relaxation rate, $\lambda$, at 0 and 0.95 GPa, for independent measurements with the initial muon spin polarisation $\mathbf{P}_\mu(0) \| c$ and $\| ab$, are shown in Fig. 2e, f; ZF data at 0.25 and 0.62 GPa are shown in Supplementary Figs. 3 and 4 in Supplementary Note 2.

To extract $T_{TRSB}$, $\lambda(T)$ is fitted with the following functional form:

$$\lambda(T) = \begin{cases} \lambda_0, & T > T_{TRSB} \\ \lambda_0 + \Delta\lambda\left[1 - \left(\frac{T}{T_{TRSB}}\right)^n\right], & T < T_{TRSB}. \end{cases} \quad (1)$$

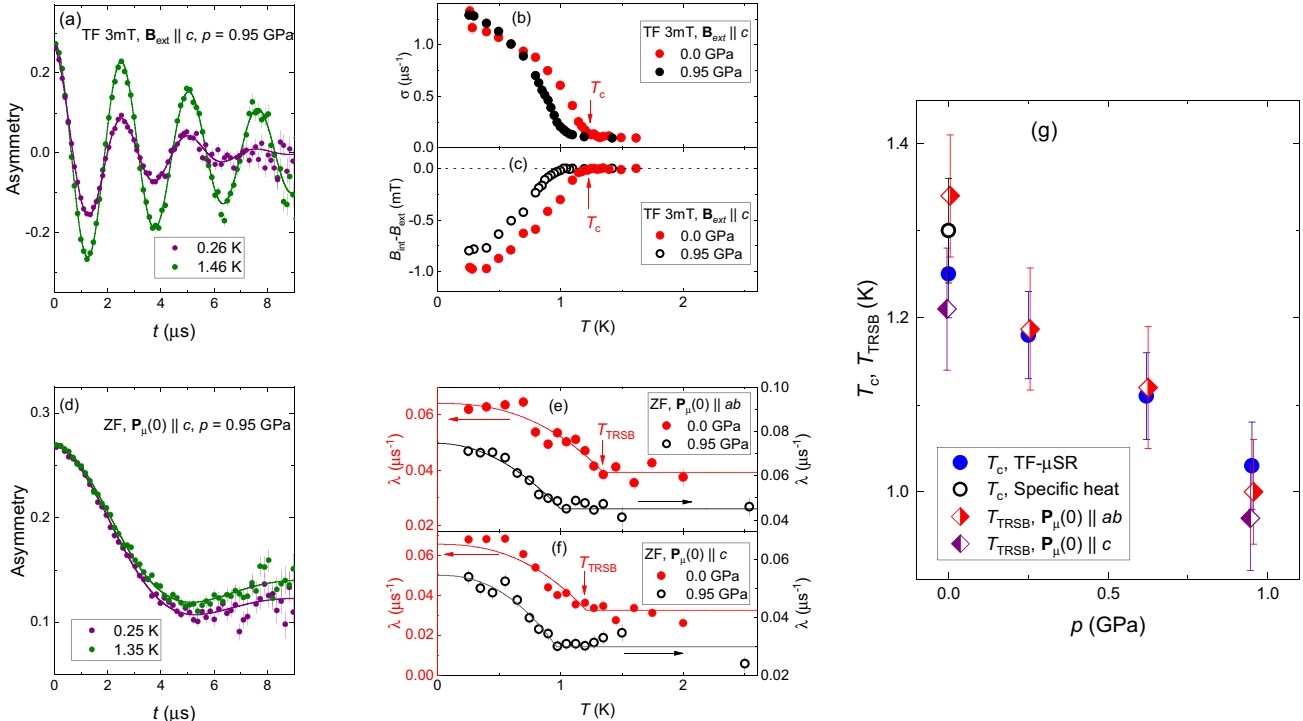

**Fig. 2 Effect of pressure on $T_c$ and $T_{TRSB}$ in Sr$_2$RuO$_4$. a** TF-$\mu$SR time-spectra above and below $T_c$ measured at $p = 0.95$ GPa and $B_{ext} = 3$ mT, with $\mathbf{B}_{ext}\|c$. The plotted quantity is the detection asymmetry between two positron detectors, which is proportional to the muon spin polarisation $P_\mu(t)$. The solid lines are fits of Eq. (2), with the sample and the pressure cell contributions described by Eqs. (3) and (4), respectively. **b, c** Temperature dependencies of the Gaussian relaxation rate $\sigma$ and the diamagnetic shift $B_{int} - B_{ext} \propto M_{FC}$ at $p = 0.0$ and 0.95 GPa. Arrows indicate the position of the superconducting transition temperature $T_c$ at $p = 0.0$ GPa. **d** ZF-$\mu$SR time-spectra above and below $T_c$, measured at $p = 0.95$ GPa and with initial muon spin polarisation $\mathbf{P}_\mu(0)\|c$. The solid lines are fits of Eq. (2), with the sample and the pressure cell parts described by Eqs. (5) and (7). **e, f** Temperature dependencies of the ZF exponential muon spin relaxation rate $\lambda$ at $p = 0.0$ and 0.95 GPa. In **e**, $\mathbf{P}_\mu(0)\|ab$, and in **f**, $\mathbf{P}_\mu(0)\|c$. The solid lines are fits of Eq. (1) to the data. Arrows indicate the position of $T_{TRSB}$ at $p = 0.0$ GPa. **g** Dependence of $T_c$ and $T_{TRSB}$ on pressure. Open circle correspond to an average $T_c$ of 1.30(6) K determined from specific heat data (see Supplementary Fig. 1 in Supplementary Note 1). The displayed error bars for $\mu$SR data correspond to one standard deviation from the $\chi^2$ fit[71]. The displayed error bars for $T_c$ indicate the rounding of the transition on a scale of approximately 0.1 K. The error bars for $\mu$SR data and $T_{TRSB}$ correspond to one standard deviation from the $\chi^2$ fit[71].

$\lambda_0$ is the relaxation rate above $T_{TRSB}$, and $\Delta\lambda$ is the enhancement due to spontaneous magnetic fields. Where data were obtained both for $\mathbf{P}_\mu(0)\|c$ and $\|ab$, the exponent $n$ is constrained to be the same for both polarisations. $T_{TRSB}$, $\lambda_0$, and $\Delta\lambda$ were obtained independently for each pressure and muon spin polarisation. The resulting values of $T_{TRSB}$ are plotted in Fig. 2g.

Our ZF data yield the following three results:

(1) Where data were taken both for $\mathbf{P}_\mu(0)\|c$ and $\|ab$ (that is, at 0 and 0.95 GPa), $T_{TRSB}$ and $\Delta\lambda$ were found to be the same within resolution for both polarisations. [At 0 GPa, $\Delta\lambda = 0.027(4)$ and $0.033(3)$ $\mu$s$^{-1}$, and at 0.95 GPa, $0.030(4)$ and $0.025(3)$ $\mu$s$^{-1}$, for $\mathbf{P}_\mu(0)\|ab$ and $\mathbf{P}_\mu(0)\|c$, respectively.] This agrees with the zero-pressure results of Luke et al.[15]. Because $\Delta\lambda$ reflects fields perpendicular to $\mathbf{P}_\mu(0)$, this result indicates that the spontaneous fields have no preferred orientation.

(2) $\Delta\lambda$ was found to be pressure-independent within resolution (including all pressures investigated: 0, 0.25, 0.62 and 0.95 GPa), having an average value of $\Delta\lambda = 0.026(2)$ $\mu$s$^{-1}$. This value corresponds to a characteristic field strength $B_{TRSB} = \Delta\lambda/\gamma_\mu = 0.031(2)$ mT. $B_{TRSB}$ has been found to vary from sample to sample[47], and this value is in line with previous reports (see refs. [15,46,48,49] and Table 1).

(3) A linear fit yields $T_{TRSB}(p) = 1.27(3)$ K $- p \cdot 0.29(5)$ K/GPa. In other words, within resolution the rate of suppression of $T_{TRSB}$ under hydrostatic pressure matches that of $T_c$.

**$\mu$SR on Sr$_{1.98}$La$_{0.02}$RuO$_4$.** Substitution of La for Sr adds electrons to the Fermi surfaces; in Sr$_{2-y}$La$_y$RuO$_4$ this doping drives the largest Fermi surface through a Lifshitz transition from an electron-like to a hole-like geometry, at $y \approx 0.20$[50,51]. At $y = 0.02$, the change in Fermi surface structure is minimal, and the main effect of the La-substitution is to suppress $T_c$, through the added disorder. Heat capacity data, measured on a small piece cut from the $\mu$SR sample, give $T_c = 0.70(5)$ K, where the error reflects the width of the transition (see Supplementary Fig. 2 in Supplementary Note 1).

This sample was studied at zero pressure. With no pressure cell material in the beam, the background is much smaller. The typical muon momentum was 28 MeV/c, giving of ~0.2 mm implantation depth[40]. Representative TF-$\mu$SR time spectra above and below $T_c$, where the applied field is $B_{ext} = 2$ mT parallel to the crystalline $c$-axis, are shown in Fig. 3a. Below $T_c$, the muon spin polarisation relaxes almost completely on a 10 $\mu$s time scale, showing that essentially the entire sample volume is super-conducting. The TF Gaussian relaxation rate $\sigma$ is shown in Fig. 3b, and $B_{int} - B_{ext}$ in Fig. 3c. These measurements yield $T_c = 0.75(5)$ K. The heat capacity data are also shown in Fig. 3b.

**Table 1 Enhancement of the exponential relaxation rates $\Delta\lambda$ and corresponding values of the spontaneous magnetic fields $B_{\text{TRSB}} = \Delta\lambda/\gamma_\mu$ caused by formation of TRSB state in $Sr_2RuO_4$ and related compounds.**

|  | $T_c$ (K) | $\Delta\lambda$ ($\mu s^{-1}$) | $B_{\text{TRSB}}$ (mT) | Reference |
|---|---|---|---|---|
| $Sr_2RuO_4$ (0.0 GPa) | 1.26 (5) | 0.030 (3) | 0.025 (3) | This study |
| $Sr_2RuO_4$ (0.25 GPa) | 1.18 (5) | 0.024 (3) | 0.021 (3) | This study |
| $Sr_2RuO_4$ (0.62 GPa) | 1.11 (5) | 0.024 (3) | 0.021 (3) | This study |
| $Sr_2RuO_4$ (0.95 GPa) | 1.03 (5) | 0.028 (3) | 0.024 (3) | This study |
| $Sr_{1.98}La_{0.02}RuO_4$ | 0.75 (5) | 0.007 (1) | 0.006 (1) | This study |
| $Sr_2RuO_4$ | $\simeq 1.45$ | $\simeq 0.037$ | $\simeq 0.032$ | ref. [15] |
| $Sr_2RuO_4$ | $\simeq 1.45$ | $\simeq 0.029$ | $\simeq 0.025$ | ref. [46] |
| $Sr_2RuO_4$ | $\simeq 1.1$ | $\simeq 0.035$ | $\simeq 0.030$ | ref. [46] |
| $Sr_2RuO_4$ | ~1.5 | $\simeq 0.041$ | $\simeq 0.035$ | ref. [48] |
| $Sr_2RuO_4$-Ru | ~1.5 | $\simeq 0.073$ | $\simeq 0.062$ | ref. [48] |
| $Sr_3Ru_2O_7$ | ~2.5 | $\simeq 0.038$ | $\simeq 0.033$ | ref. [48] |
| $Sr_2RuO_4$ | $\simeq 1.45$ | $\simeq 0.020$ | $\simeq 0.017$ | ref. [49] |
| $Sr_2RuO_4$ | 1.38 (4) | 0.0088 (10) | 0.0075 (9) | ref. [47] |
| $Sr_2RuO_4$ | 1.22 (6) | 0.024 (2) | 0.020 (2) | ref. [47] |

Only the experiments with preserved tetragonal lattice symmetry resulting in $T_c \simeq T_{\text{TRSB}}$ are considered. In a case when the results of both, $\mathbf{P}_\mu(0)\|c$ and $\mathbf{P}_\mu(0)\|ab$ experiments are available (present study and ref. [15]), the values of $\Delta\lambda$ are averaged out.

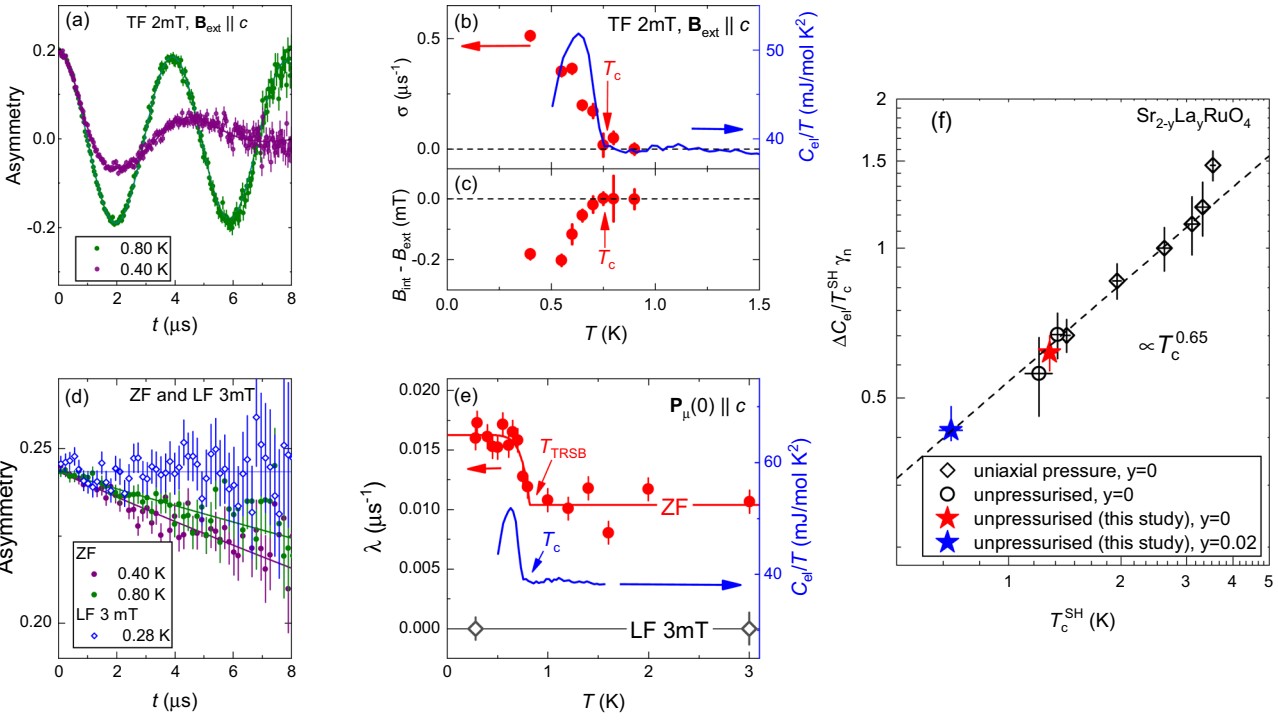

**Fig. 3 TRSB in $Sr_{1.98}La_{0.02}RuO_4$. a** TF-$\mu$SR time-spectra above and below $T_c$ measured at $B_{\text{ext}} = 2$ mT with $\mathbf{B}_{\text{ext}}\|c$. The solid lines are fits of Eq. (3) to the data. **b**, **c** Temperature dependencies of the Gaussian relaxation rate $\sigma$ and the diamagnetic shift $B_{\text{int}} - B_{\text{ext}}$, respectively. Arrows indicates the superconducting transition temperature $T_c$, determined from the TF-$\mu$SR data. The blue curve in **b** is the electronic specific heat $C_{\text{el}}/T$, measured on a small piece cut from the $\mu$SR sample. **d** ZF- and LF-$\mu$SR time-spectra. ZF data from above and below $T_c$, measured with $\mathbf{P}_\mu(0)\|c$, are shown. The LF data are from $T$ well below $T_c$, and with $\mathbf{B}_{\text{ext}} = 3$ mT $\|\mathbf{P}_\mu(0)$. The solid lines are fits of Eq. (8). **e** Temperature dependence of the ZF and LF exponential relaxation rate $\lambda$. The solid red line is the fit of Eq. (1) to ZF $\lambda(T)$ data. The blue curve is, again, $C_{\text{el}}/T$. Arrows indicates positions of $T_c$ and $T_{\text{TRSB}}$. **f** Double logarithmic plot of the normalised specific heat jump $\Delta C_{\text{el}}/\gamma_n T_c^{\text{SH}}$ versus $T_c^{\text{SH}}$ [$\gamma_n$ is the Sommerfeld coefficient and $T_c^{\text{SH}}$ is the transition temperature determined from $C_{\text{el}}/T(T)$ by means of equal-entropy construction, see Supplementary Fig. 1 in Supplementary Note 1]. Values of $\Delta C_{\text{el}}/\gamma_n T_c^{\text{SH}}$ are determined in a way presented in Supplementary Fig. 2 in Supplementary Note 1. Filled symbols: data from this work; open symbols: data taken from refs. [47,52]. The displayed error bars for $\mu$SR data correspond to one standard deviation from the $\chi^2$ fit[71]. The error bars for $\Delta C_{\text{el}}/\gamma_n T_c^{\text{SH}}$ and $T_c^{\text{SH}}$ indicate uncertainty in selecting the temperature range for linear fit below $T_c$.

ZF-$\mu$SR data are presented in Fig. 3d, e. Fitting with Eq. (1) returns $\Delta\lambda = 0.007(1)$ $\mu s^{-1}$ and $T_{\text{TRSB}} = 0.8(1)$ K. This $\Delta\lambda$ is noticeably smaller than that obtained from the undoped $Sr_2RuO_4$ sample, corresponding to an internal field $B_{\text{TRSB}} \approx 0.01$ mT. It is,

however, within the range of previous results[47]. In qualitative agreement with data on a lower $T_c$ $Sr_2RuO_4$, reported in ref. [46], though here with more data at $T > T_c$ to be certain of the base relaxation rate, this low value of $\Delta\lambda$ shows that $B_{\text{TRSB}}$ is not

straightforwardly related to defect density. At present, the origin of the sample-to-sample variation in $B_{TRSB}$ is unknown.

Longitudinal-field (LF) measurements can be employed to determine whether internal fields are static or fluctuating. If $B_{TRSB}$ is static, under an applied field parallel to $\mathbf{P}_\mu(0)$ that is considerably larger than $B_{TRSB}$, muon spin precession is greatly restricted and the spin polarisation does not relax (i.e. the muon spins decouple from $B_{TRSB}$). In contrast, fluctuating $B_{TRSB}$ can still relax the muon spin polarisation[40]. Data shown in Fig. 3d, e indicate that $\mathbf{B}_{ext}||\mathbf{P}_\mu(0) = 3$ mT fully suppresses the muon spin relaxation, and therefore that $B_{TRSB}$ is static on a microsecond time scale, in agreement with data on clean $Sr_2RuO_4$ reported in ref. [15]. We note that LF measurements were not performed on the hydrostatically pressurised sample because the decoupling field for the Cu background is of the order of 10 mT, considerably stronger than that for $Sr_2RuO_4$.

**Heat capacity measurements**. The specific heat measurements were performed at ambient pressure for several pieces of $Sr_{2-y}La_yRuO_4$ single crystals. The results are presented in Fig. 3b, e for $Sr_{1.98}La_{0.02}RuO_4$ ($y = 0.02$) and in Supplementary Fig. 1 in Supplementary Note 1 for $Sr_2RuO_4$ ($y = 0.0$), respectively. The specific heat jumps at $T_c$ ($\Delta C_{el}/\gamma_n T_c$, $\gamma_n$ is the Sommerfeld coefficient) were further obtained in a way presented in Supplementary Fig. 2 in Supplementary Note 1.

Figure 3f summarises the $\Delta C_{el}/\gamma_n T_c^{SH}$ vs. $T_c^{SH}$ data for our $Sr_{2-y}La_yRuO_4$ samples. Here $T_c^{SH}$ denotes the superconducting transition temperature determined from $C_{el}/T$ vs. $T$ measurement curves by means of equal-entropy construction algorithm, see Supplementary Fig. 1a in Supplementary Note 1. In addition, we have also included some literature data for $Sr_2RuO_4$ with different amount of disorder[47], and for $Sr_2RuO_4$ under uniaxial strain[52]. In total, Fig. 3f compares $Sr_2RuO_4$ samples with a factor of five variation in $T_c$. Remarkably, $\Delta C_{el}/\gamma_n T_c$ vs. $T_c$ data points scale as $T_c^\alpha$ with $\alpha \approx 0.65$, which is distinctly different from the BCS behaviour, where $\alpha = 0$ ($\Delta C_{el}/\gamma_n T_c = const$). Just a single point at $T_c \simeq 3.5$ K deviates from the scaling behaviour, which might be associated with tuning the electronic structure of $Sr_2RuO_4$ close to a van Hove singularity[52]. The results presented in Fig. 3f indicate, therefore, that the perturbation changes the gap structure on the Fermi surface, i.e. its "anisotropy" or the distribution among the three different bands which can lead to a renormalisation of the specific heat jump being not simply proportional to the normal-state-specific heat above $T_c$.

Such scaling behaviour is rarely observed since the ratio $\Delta C_{el}/\gamma_n T_c$ is sensitive to a change of the superconducting gap structure and symmetry. Note that a similar scaling is reported for Fe-based superconductors, where $\Delta C_{el}/\gamma_n T_c$ follows approximately the BNC (Bud'ko-Ni-Canfield) scaling behaviour $\Delta C_{el}/\gamma_n T_c \propto T_c^\alpha$ with $\alpha \approx 2$[53], which is considered to be a consequence of the unconventional multiband $s \pm$ superconductivity. The change of the superconducting pairing state in the $Ba_{1-x}K_xFe_2As_2$ system results in abrupt change of the scaling behaviour leading to an intermediate $s + is$ state[11]. The monotonic $\Delta C_{el}/\gamma_n T_c$ vs. $T_c$ behaviour obtained in the present study suggests, therefore, that La-substitution do not yield a change of the superconducting gap symmetry. Consequently, the superconducting gap structure does not undergo a significant change due to effects of disorder and it remains the same as in bare $Sr_2RuO_4$ compound.

## Discussion

In a previous ZF-$\mu$SR experiment, in-plane uniaxial pressure, which does lift the tetragonal symmetry of the unpressurised lattice, was found to induce a strong splitting between $T_c$ and $T_{TRSB}$[47]. Uniaxial pressure drives a strong increase in $T_c$, while

$T_{TRSB}$ varies much more weakly, probably decreasing slightly with initial application of pressure. The microscopic mechanism yielding the signal observed at $T_{TRSB}$, a weak enhancement in muon spin relaxation rate, remains unclear: the main proposed mechanism, magnetism induced at defects and domain walls by a TRSB superconducting order, is unproved experimentally[54,55]. At present, the link between enhanced muon spin relaxation and TRSB superconductivity is, therefore, mainly empirical, based on: (1) the facts that it is a signal seen in only a small fraction of known superconductors, (2) it generally appears at $T_c$ and (3) the general notion that TRSB superconductivity can in principle generate magnetic fields, while muons detect magnetic fields. In ref. [47], careful checks were performed to rule out instrumentation artefact as the origin of the signal at $T_{TRSB}$, and it was further argued that this signal is extremely difficult to obtain from a purely magnetic mechanism. Nevertheless, the weak observed variation of $T_{TRSB}$, while $T_c$ varied strongly, raised some doubt as to whether this signal is in fact associated with the superconductivity.

Here, we have observed a clear suppression of $T_{TRSB}$ with hydrostatic stress, at a rate matching the suppression of $T_c$. This result further strengthens the evidence that enhanced muon spin relaxation is an indicator of TRSB superconductivity: $T_{TRSB}$ tracks $T_c$ when tetragonal lattice symmetry is preserved, while the splitting induced by uniaxial pressure shows unambiguously that it is a distinct transition, and not an artefact through some unidentified mechanism of the superconducting transition itself. Figure 4 shows $T_{TRSB}$ versus $T_c$. The data reported here, on hydrostatically pressurised $Sr_2RuO_4$ and on unpressurised $Sr_{1.98}La_{0.02}RuO_4$, fall on the $T_{TRSB} = T_c$ line, while the uniaxial pressure data from ref. [47] clearly deviate from this line.

Our central finding that $T_{TRSB}$ tracks $T_c$ provides further support for the single-representation $d_{xz} \pm id_{yz}$ order parameter. Importantly, homogeneous $d_{xz} \pm id_{yz}$ is the only spin-singlet order parameter consistent with the selection rules imposed by ultrasound and Kerr effect data. Ultrasound data on $Sr_2RuO_4$[56,57] show a type of renormalisation that is not possible for a single-component order parameter on a tetragonal lattice: a jump in ultrasound velocity at $T_c$ for transverse modes. While these experimental results are not sensitive to the spin configuration, they impose other stringent conditions on the possible pairing symmetries[58,59]. The polar Kerr effect mentioned above is a second experiment which provides symmetry-related constraints, being compatible only with chiral pairing states[16]. These two selection rules are obeyed by both the chiral $p$-wave and chiral $d$-wave state, though as noted in the Introduction, $p$-wave order appears to be ruled out by NMR Knight shift data[24,25]. In contrast, the composite-representation states do not satisfy the requirements for both selection rules. The $d_{x^2-y^2} + ig_{xy(x^2-y^2)}$ and $s + id_{xy}$ states are constructed to be compatible with the ultrasound measurements, but they are not chiral[31,60]. The $s + id_{x^2-y^2}$ state violates both selection rules[29]. It can be generally stated that any composite-representation pairing states in a tetragonal crystal, composed of components of two one-dimensional representations, would satisfy at most one of the two selection rules (see the "Methods" section).

We note that there is a recent proposal for inhomogeneous superconductivity in $Sr_2RuO_4$: single-component ($d_{x^2-y^2}$) in the bulk, but two-component ($d_{x^2-y^2} + ig_{xy(x^2-y^2)}$) in the strain fields around dislocations[61]. The combination of phase locking between adjacent dislocations and a preferred orientation to the dislocations would result in a bulk chirality. In this proposal, $T_{TRSB}$ could be locked to $T_c$ by hypothesising that superconductivity appears at the dislocations before the bulk, but tuning would then

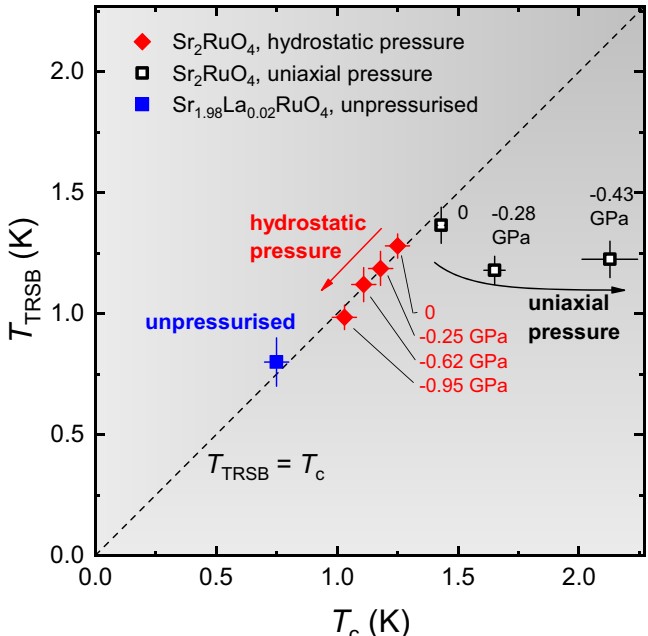

**Fig. 4 Relation between $T_{\text{TRSB}}$ and $T_c$.** Dependence of the time reversal symmetry breaking temperature $T_{\text{TRSB}}$ on the superconducting transition temperature $T_c$. The closed symbols correspond to the results obtained in present studies under hydrostatic pressure up to 0.95 GPa in pure $Sr_2RuO_4$ (diamonds) and in the La-doped $Sr_{2-y}La_yRuO_4$ with $T_c = 0.75(5)$ K (square). The open squares are the uniaxial pressure data for undoped $Sr_2RuO_4$ from ref. [47]. The dashed line corresponds to $T_{\text{TRSB}} = T_c$. The minus signs at the pressure values denote the effect of 'compression' of the sample volume. The error bars are the same as defined in Figs. 2 and 3 and in ref. [47].

be required to obtain $T_c$ and $T_{\text{TRSB}}$ that split under modest uniaxial stress.

Major challenges to $d_{xz} \pm id_{yz}$ order are the absence of a resolvable second heat capacity anomaly at $T_{\text{TRSB}}$ in measurements on uniaxially pressurised $Sr_2RuO_4$[52], and, as already noted, the theoretical challenges in obtaining a horizontal line node in a highly two-dimensional metal[62]. We note in addition that an analysis of low-temperature thermal conductivity data indicated vertical, rather than horizontal, line nodes in $Sr_2RuO_4$[63]. The theoretical objection to horizontal line nodes might be overcome through the complex nature of the multi-orbital band structure, including sizable spin-orbit coupling[26,64,65].

So we may conclude that our ZF-μSR data combined with the selection rules for ultrasound and polar Kerr effect and the NMR Knight shift behaviour are consistent with the single-representation chiral $d_{xz} + id_{zy}$-wave state, while all composite-representation states suffer from several deficiencies. We note, however, that there are also empirical challenges to a hypothesis of $d_{xz} \pm id_{yz}$, and that the difficulty in reconciling apparently contradictory experimental results in $Sr_2RuO_4$ may mean that one or more major, apparently solid results will in time be found to be incorrect, either for a technical reason or in interpretation. Further experiments are therefore necessary.

## Methods

**$Sr_{2-y}La_yRuO_4$ single crystals.** Single crystals of $Sr_{2-y}La_yRuO_4$ were grown by means of a floating zone technique[39]. Samples for measurement under hydrostatic pressure (with $y = 0$) were cut from two rods, C140 and C171, that each grew along a ⟨100⟩ crystallographic direction. The rods have diameter $\varnothing \simeq 3$ mm. Two sections of length 8–12 mm were taken from each rod. These were then cleaved, forming semi-cylindrical samples with flat surfaces perpendicular to the $c$-axis (see Fig. 1a).

The effect of La doping on the TRSB transition was studied on a single original $Sr_{2-y}La_yRuO_4$ crystal of length 8 mm. The La concentration was analysed by an electron-probe micro-analysis and was found to be $y \simeq 0.02$. Before the μSR measurements, this rod was then cleaved into two semi-cylindrical pieces, again with the flat faces ⊥$c$.

The x-ray diffraction experiments performed on small powdered pieces cut from of each particular rod gave $a = 0.3867$ nm, $c = 1.273$ nm for pure $Sr_2RuO_4$ and $a = 0.3865$ nm, $c = 1.274$ nm for La-substituted sample.

**Specific heat of $Sr_{2-y}La_yRuO_4$ at ambient pressure.** Specific heat measurements were performed at zero pressure for several pieces of $Sr_{2-y}La_yRuO_4$ single crystals, cut from the rod used for μSR measurements.

For $Sr_2RuO_4$ used in hydrostatic pressure measurements, the electronic specific heat capacity $C_{el}/T$ was measured for four samples: one sample cut from each end of both the C140 and C171 sections. Results are presented in Supplementary Fig. 1 in Supplementary Note 1.

The temperature dependence of $C_{el}/T$ for a small piece cut from the $Sr_{1.98}La_{0.02}RuO_4$ μSR sample is presented in Fig. 3b, e and Supplementary Fig. 2 in Supplementary Note 1.

**μSR experiments and μSR data analysis procedure.** The muon spin rotation/relaxation (μSR) experiments were performed at the μE1 and πE1 beamlines, using the GPD[35], and Dolly spectrometers (Paul Scherrer Institute, PSI Villigen, Switzerland). At the GPD instrument, experiments under pressure up to $p \simeq 0.95$ GPa on undoped $Sr_2RuO_4$ were performed. At the Dolly spectrometer, measurements of $Sr_{1.98}La_{0.02}RuO_4$ at ambient pressure were conducted. At both instruments $^4$He cryostats equipped with the $^3$He insets (base temperature $T \simeq 0.25$ K) were used.

At the GPD instrument, measurements in zero-field (ZF-μSR) and with the field applied transverse to the initial muon spin polarisation $\mathbf{P}_\mu(0)$ (TF-μSR) were performed. In two sets of ZF-μSR studies, $\mathbf{P}_\mu(0)$ was set to be parallel to the $c$-axis and along the $ab$-plane, respectively. In TF-μSR measurements the small 3 mT magnetic field was applied parallel to the $c$-axis and perpendicular to $\mathbf{P}_\mu(0)$.

At the Dolly instrument, in addition to ZF- and TF-μSR experiments, the LF measurements were performed. In these studies 3 mT magnetic field was applied parallel to the $c$-axis and to the initial muon spin polarisation $\mathbf{P}_\mu(0)$.

The experimental data were analysed by separating the μSR signal on the sample (s) and the background (bg) contributions[66]:

$$A_0P(t) = A_sP_s(t) + A_{bg}P_{bg}(t). \quad (2)$$

Here $A_0$ is the initial asymmetry of the muon spin ensemble, and $A_s$ ($A_{bg}$) and $P_s(t)$ [$P_{bg}(t)$] are the asymmetry and the time evolution of the muon spin polarisation for muons stopped inside the sample (outside of the sample), respectively.

In a case of μSR under pressure studies, the background contribution (~50% of total μSR response) is determined by the muons stopped in the pressure cell body. At ambient pressure experiment the small background contribution (of the order of 5%) is caused by muons stopped in the sample holder and the cryostat windows.

In TF-μSR experiments, the sample contribution was analysed by using the following functional form:

$$P_s^{TF}(t) = \exp\left[-\frac{\sigma^2 t^2}{2}\right]\cos(\gamma_\mu B_{int}t + \phi). \quad (3)$$

Here $B_{int}$ is the internal field in the sample, $\phi$ is the initial phase of the muon spin ensemble, and $\gamma_\mu \simeq 2\pi \times 135.5$ MHz/T is the muon gyromagnetic ratio. The Gaussian relaxation rate $\sigma$ consists of the "superconducting", $\sigma_{sc}$, and nuclear moment, $\sigma_{nm}$, contributions and it is defined as: $\sigma^2 = \sigma_{sc}^2 + \sigma_{nm}^2$. Here, $\sigma_{sc}$ and $\sigma_{nm}$ characterise the damping due to the formation of the flux-line lattice in the superconducting state and of the nuclear magnetic dipolar contribution, respectively. In the analysis, $\sigma_{nm}$ was assumed to be constant over the entire temperature range and was fixed to the value obtained above $T_c$, where only nuclear magnetic moments contribute to the muon depolarisation rate (see Supplementary Fig. 3a in Supplementary Note 2).

The pressure cell contribution was described by using the following equation:

$$P_{pc}^{TF}(t) = \exp\left[-\frac{\sigma_{pc}^2 t^2}{2}\right]\cos(\gamma_\mu B_{ext}t + \phi). \quad (4)$$

Here $\sigma_{pc} \simeq 0.28$ μs$^{-1}$ is the field and the temperature-independent relaxation rate of beryllium-copper[35], and $B_{ext}$ is the externally applied field.

The solid lines in Fig. 2a correspond to the fit of TF-μSR data by using Eq. (2) with the sample and the background parts described by Eqs. (3) and (4). For the data presented in Fig. 3a the background contribution was described by non-relaxing function $P_{bg}^{TF}(t) = \cos(\gamma_\mu B_{ext}t + \phi)$. The good agreement between the fits and the data demonstrates that the above model describes the experimental data rather well.

With the external magnetic field applied along the crystallographic $c$-axis ($\mathbf{B}_{ext}\|c$), the superconductig contribution into the Gaussian relaxation rate $\sigma_{sc}$ becomes proportional to the inverse squared in-plane magnetic penetration depth $\lambda_{ab}^{-2}$[42]. The proportionality coefficient between $\sigma_{sc}$ and $\lambda_{ab}^{-2}$ depends on the value of

the applied field, the symmetry of the flux-line lattice and the angular dependence of the superconducting order parameter.

The temperature dependencies of the Gaussian relaxation rate $\sigma$ and the diamagnetic shift $B_{int} - B_{ext}$ are presented in Figs. 2b, c and 3b, c for $Sr_2RuO_4$ and $Sr_{1.98}La_{0.02}RuO_4$ samples, respectively.

In ZF- and LF-$\mu$SR experiments the sample contribution includes both, the nuclear moment relaxation and an additional exponential relaxation $\lambda$ caused by appearance of spontaneous magnetic fields[15]:

$$P_s^{ZF}(t) = GKT_s(t)\,e^{-\lambda t}. \tag{5}$$

Here $GKT(t)$ is the Gaussian Kubo-Toyabe (GKT) relaxation function describing the magnetic field distribution created by the nuclear magnetic moments[40,67]:

$$GKT(t) = \frac{1}{3} + \frac{2}{3}(1 - \sigma_{GKT}^2 t^2)\,e^{-\sigma_{GKT}^2 t^2/2}. \tag{6}$$

$\sigma_{GKT}$ is the GKT relaxation rate.

Muons implanted in beryllium-copper pressure cell body sense solely the magnetic field distribution created by copper nuclear magnetic moments and described as:

$$P_{pc}^{ZF}(t) = GKT_{pc}(t) \tag{7}$$

with the temperature-independent relaxation rate $\sigma_{GKT,BeCu} \simeq 0.35\ \mu s^{-1}$[35].

Fits of Eq. (2), with the sample and pressure cell parts described by Eqs. (5) and (7), to the ZF-$\mu$SR data were performed globally. The ZF-$\mu$SR time-spectra taken at each particular muon spin polarisation [$\mathbf{P}_\mu(0) \| ab$ and $\mathbf{P}_\mu(0) \| c$] and pressure ($p = 0.0$, 0.25, 0.62 and 0.95 GPa) were fitted simultaneously with $A_s$, $A_{pc}$, $\sigma_{GKT,Sr_2RuO_4}$, $\sigma_{GKT,BeCu}$, and $\lambda_0$ as common parameters, and $\lambda$ as individual parameter for each particular data set. The solid green and purple lines in Fig. 2d correspond to the fit of ZF-$\mu$SR data by using Eq. (2) with the sample and the background parts described by Eqs. (5) and (7).

Note that the absence of strong nuclear magnetic moments in $Sr_{2-y}La_yRuO_4$ leads to the corresponding Gaussian Kubo-Toyabe relaxation rate being nearly zero. Consequently, the analysis of ZF- and LF-$\mu$SR data for $Sr_{1.98}La_{0.02}RuO_4$ was performed by using the simple-exponential decay function:

$$P_s^{ZF,LF}(t) = e^{-\lambda t}. \tag{8}$$

The solid lines in Fig. 3d correspond to the fit of ZF-$\mu$SR data by using Eq. (2) with the sample part described by Eq. (8) and the non-relaxing background $P_{bg}^{ZF,LF}(t) = 1$.

The temperature dependencies of the exponential relaxation rate $\lambda$ are presented in Figs. 2e, f and 3e for $Sr_2RuO_4$ and $Sr_{1.98}La_{0.02}RuO_4$ samples, respectively.

**Symmetry properties of the order parameters.** Several order parameters have been proposed for the time reversal symmetry breaking superconducting state of $Sr_2RuO_4$. We would like here to give a brief overview on the different options and the symmetry requirements to satisfy the selection rules for two experiments: ultrasound velocity renormalisation for the transverse $c_{66}$-mode and the polar Kerr effect. For tetragonal crystal symmetry with the point group $D_{4h}$ the even parity spin-singlet pairing states can be listed according to the irreducible representations of $D_{4h}$, four one-dimensional ones $A_{1g}$, $A_{2g}$, $B_{1g}$, $B_{2g}$ and a two-dimensional one $E_u$. The pair wave function $\psi_\Gamma(\boldsymbol{k})$ of the corresponding states are given by:

$$
\begin{aligned}
\psi_{A_{1g}}(\boldsymbol{k}) &= \psi_0(\boldsymbol{k}) & s\text{-wave} \\
\psi_{A_{2g}}(\boldsymbol{k}) &= \psi_0(\boldsymbol{k})k_x k_y(k_x^2 - k_y^2) & g_{xy(x^2-y^2)}\text{-wave} \\
\psi_{B_{1g}}(\boldsymbol{k}) &= \psi_0(\boldsymbol{k})(k_x^2 - k_y^2) & d_{x^2-y^2}\text{-wave} \\
\psi_{B_{2g}}(\boldsymbol{k}) &= \psi_0(\boldsymbol{k})k_x k_y & d_{xy}\text{-wave} \\
\psi_{E_g}(\boldsymbol{k}) &= \{\psi_0(\boldsymbol{k})k_x k_z, \psi_0(\boldsymbol{k})k_y k_z\} & \{d_{xz}, d_{yz}\}\text{-wave}
\end{aligned}
\tag{9}
$$

where $\psi_0(\boldsymbol{k})$ is a function of $\boldsymbol{k}$ invariant under all symmetry operations of the tetragonal lattice. We list here first the composite-representation TRSB states:

$$
\begin{aligned}
\tilde\Gamma_1 &= A_{1g} \oplus A_{2g}: & s + ig\text{-wave} \\
\tilde\Gamma_2 &= A_{1g} \oplus B_{1g}: & s + id\text{-wave} \\
\tilde\Gamma_3 &= A_{1g} \oplus B_{2g}: & s + id'\text{-wave} \\
\tilde\Gamma_4 &= B_{1g} \oplus A_{2g}: & d + ig\text{-wave} \\
\tilde\Gamma_5 &= B_{2g} \oplus A_{2g}: & d' + ig\text{-wave} \\
\tilde\Gamma_6 &= B_{1g} \oplus B_{2g}: & d + id'\text{-wave}
\end{aligned}
\tag{10}
$$

Note that in general different representations correspond to different critical temperature. Thus, to obtain a single superconducting phase transition for the composite states an accidental degeneracy of two representations is necessary. The two states proposed so far are $\tilde\Gamma_2$[29,30] and $\tilde\Gamma_4$[31,32]. The two-dimensional representation allows for the combination:

$$\tilde\Gamma_7 = E_g: \quad \text{chiral } d\text{-wave} \tag{11}$$

with a pair wave function $\psi_{E_g}(\boldsymbol{k}) = \psi_0(\boldsymbol{k})k_z(k_x \pm ik_y)$ as proposed in refs. [26,62]. All

composite states, $\tilde\Gamma_{1-6}$, can be constructed by electron pairing within the $RuO_2$ planes, while the state $\tilde\Gamma_7$ requires interlayer pairing. Due to the spin-singlet nature all states are compatible with the new NMR Knight shift results[24,25]. All TRSB state are expected to generate internal spontaneous currents around defects, such as surfaces and domain walls and, consequently, under present understanding are compatible with the $\mu$SR experiments[15].

Next we consider the two selection rules. For the coupling to the lattice we restrict consideration to the mode which corresponds to the elastic constant $c_{66}$, which is connected with the strain tensor element $\epsilon_{xy} = \epsilon_{yx}$[58,59]. This is active for transverse modes with a wave vector in the plane, e.g. [100] and a polarisation perpendicular also within the plane. This strain tensor component belongs by symmetry to the representation $B_{2g}$[58,59,68]. For the observed renormalisation of the speed of sound the superconducting order parameter has to couple linearly to $\epsilon_{xy}$, thus, requiring that $B_{2g}$ is contained in the decomposition of $\tilde\Gamma_j \otimes \tilde\Gamma_j$. This only possible for $\tilde\Gamma_3$, $\tilde\Gamma_4$ and $\tilde\Gamma_7$:

$$\tilde\Gamma_3 \otimes \tilde\Gamma_3 = \tilde\Gamma_4 \otimes \tilde\Gamma_4 = 2A_{1g} \oplus 2B_{2g} \tag{12}$$

and

$$\tilde\Gamma_7 \otimes \tilde\Gamma_7 = A_{1g} \oplus A_{2g} \oplus B_{1g} \oplus B_{2g}. \tag{13}$$

The selection rule resulting in the polar Kerr effect requires the order parameter to couple by symmetry to the $z$-component of the magnetic field, $B_z$ which belongs to the representation $A_{2g}$. Again we consider the decomposition of the corresponding representations of the different pairing states. We find that only $\tilde\Gamma_1$, $\tilde\Gamma_6$ and $\tilde\Gamma_7$ satisfy the condition. The only pairing state which appears to obey both selection rules is the chiral $d$-wave state. None of the composite pairing states can satisfy both conditions. Among them there are the states $\tilde\Gamma_2$ and $\tilde\Gamma_5$ which are in conflict with both selection rules.

Turning to the odd parity states the analogous picture arises with:

$$
\begin{aligned}
\boldsymbol{d}_{A_{1u}}(\boldsymbol{k}) &= \psi_0(\boldsymbol{k})(\hat{x}k_x + \hat{y}k_y) \\
\boldsymbol{d}_{A_{2u}}(\boldsymbol{k}) &= \psi_0(\boldsymbol{k})(\hat{x}k_y - \hat{y}k_x) \\
\boldsymbol{d}_{B_{1u}}(\boldsymbol{k}) &= \psi_0(\boldsymbol{k})(\hat{x}k_x - \hat{y}k_y) \\
\boldsymbol{d}_{B_{2u}}(\boldsymbol{k}) &= \psi_0(\boldsymbol{k})(\hat{x}k_y + \hat{y}k_x) \\
\boldsymbol{d}_{E_u}(\boldsymbol{k}) &= \psi_0(\boldsymbol{k})\{\hat{z}k_x, \hat{z}k_y\}.
\end{aligned}
\tag{14}
$$

here listed in the convenient $\boldsymbol{d}$-vector notation for spin-triplet pairing states (see ref. [68]). It is important to note that all composite phases from combination of two pairing states of one-dimensional representation are $c$-axis equal spin state and would be in agreement with present time NMR Knight data[24,25] and had been proposed as possible states in refs. [69,70]. These states are also called helical state in literature, as they are topologically non-trivial with helical surface states. The Knight shift experiments disagree with expectations of the state in representation $E_u$ which yields the chiral $p$-wave state.

Again we have to make composite states of the one-dimensional representation to obtain TRSB phases. Analogous to the even parity case we do not find any composite state which satisfies both selection rules, in contrast to the chiral $p$-wave state which behaves the same way as the chiral $d$-wave state in this respect.

## Data availability

All data needed to evaluate the conclusions in the paper are present in the paper and/or in the Supplementary Information. Other data that support the plots within this paper and other findings of this study are available from the corresponding author upon reasonable request.

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

## Acknowledgements
The work was performed at the Swiss Muon Source (S$\mu$S), Paul Scherrer Institute (PSI, Switzerland). We acknowledge fruitful discussions with Zurab Guguchia, Carsten Timm and Jing Xia. Matthias Elender is acknowledged for technical support. The work of R.G. is supported by the Swiss National Science Foundation (SNF Grant No. 200021-175935). The work of M.S and B.Z was financially supported by the Swiss National Science Foundation (SNSF) through Division II (Grant No. 184739). The work of V.G. was supported by DFG GR 4667/1. N.K. acknowledges the support from JSPS KAKENHI (Nos. JP18K04715, and JP21H01033) in Japan. Y.M. acknowledges funding by JPJS-CNR-SPIN Core-to-core programme (No. JPJSCCA20170002), and by JPJS KAKENHI Nos. JP15H05852, JP15K21717, and JP17H06136. The work of H.-H.K. was supported by DFG SFB 1143 and GRK 1621.

## Author contributions
R.K., V.G. and M.S. conceived the project. Data were taken by R.K., V.G., D.D. and R.G. R.K. and V.G. performed data analysis and interpreted the results together with M.S. B.Z. and M.S. provided the theoretical analysis. N.K. provided and characterised samples. R.K., V.G., M.S. and C.W.H. wrote the manuscript with inputs from all authors: D.D., R.G., B.Z., N.K., Y.M. and H.-H.K.

## Competing interests
The authors declare no competing interests.
