## [Peer Review File · Nature Communications]

REVIEWER COMMENTS

Reviewer #1 (Remarks to the Author):

The manuscript "Unsplit superconducting and time reversal symmetry breaking transitions in Sr_2RuO_4 under hydrostatic pressure and disorder." by Dr Khasanov and colleagues is perfect and can serve as a text book example for a good scientific paper. It introduces a clear question (what is the order parameter type in SRO), explains the tools used to address the question (critical temperatures measurements under pressure), introduces tools that are not trivial and specially designed to address the question (high pressure μSR), the answer from the experiment is clear (the critical temperatures are proportional), and finally, the conclusions are not over-selling and point to the remaining difficulties. The subject matter of the manuscript draws a lot of attention. The manuscript is very well written and I could barely find any typos. I therefore strongly support publication of the manuscript in Nature Communications.

The only problem is that time reversal symmetry breaking signals of superconductors are always so weak that they are on the edge of the sensitivity capability. That does not mean that they are not there, but the discussion is not over.

Minor comments are:

While the reason for measuring the relation between the two critical temperatures, T_{TRSB} and T_c , is very well explained, the motivation for measuring specific heat is not covered. Consequently, the specific heat data seems somewhat out of place, as if it is introduced into the paper just because the authors had the data available.

The quantity B_{ext} is not well defined. Is it the field given by the power supply or a measurement above T_c ?

The caption of Fig. ED1 contains the non-existing word "minim".

In Eq. M8, $d_{\{x^2-y^2\}}$, brackets are missing around $(k_x^2-k_y^2)$.

Reviewer #2 (Remarks to the Author):

This paper makes an important claim about the order parameter of Sr_2RuO_4 , providing a direct test of the symmetry on the basis of whether the TRSB temperature tracks T_c or not. The conclusion points to $d_{\{xz\}} \pm id_{\{yz\}}$ symmetry rather than $s+id$ or related possibilities. This has been achieved by using hydrostatic pressure (which does not break the tetragonal symmetry) and also La doping (a possibly less clean probe, but which also does not break the tetragonal symmetry - and has the added advantage that the measured signal is not dominated by the background from a pressure cell). The research has been carried out to a high degree of quality and the analysis has been done well. I am persuaded by the result and recommend publication. The topic is of very high current interest given the renewed attention paid to Sr_2RuO_4 since the original NMR triplet superconductivity study was disproven. Therefore this paper will attract attention.

The authors do not mention any x-ray characterisation of their La-doped crystals. Was this done? How much do the lattice parameters vary?

The main result seems quite clear but the sample-to-sample variation in B_{TRSB} , as deduced from $\Delta\lambda$ is rather concerning. I appreciate that the authors state that they do not have a good explanation for this, but it might be helpful for future studies if they could provide a table of measured values of this parameter from their and other previously published studies to demonstrate the effect and provide a pointer for future work to explain this. Of course the problem with the usual group theory arguments is they can only tell you whether an effect is possible or forbidden, not what its strength is < if > it is allowed.

Response to Referee #1

We are grateful to the Referee for recommending our paper for publication.

Referee: While the reason for measuring the relation between the two critical temperatures, T_{TRSB} and T_c , is very well explained, the motivation for measuring specific heat is not covered. Consequently, the specific heat data seems somewhat out of place, as if it is introduced into the paper just because the authors had the data available.

Answer: The results of the specific heat measurements are intended to show that the perturbations, like the La-substitution and/or uniaxial pressures do not affect the superconducting gap structure of Sr₂RuO₄. In the revised version of the manuscript we have added: "The monotonic $\Delta C_{el} / \gamma_n T_c$ vs. T_c behavior obtained in the present study suggests, therefore, that La-substitution, do not yield a strong qualitative change of the superconducting gap symmetry. Consequently, the superconducting gap structure does not undergo a significant change due to effects of disorder and it remains the same as in bare Sr₂RuO₄ compound."

Referee: The quantity B_{ext} is not well defined. Is it the field given by the power supply or a measurement above T_c ?

Answer: The external field at the sample position is generated by a Helmholtz coil system. The value of the field as a function of applied current was initially calibrated at all μSR instruments by measuring the muon-spin precession frequency $\omega = \gamma \mu B_{\text{ext}}$ on the paramagnetic sample (metallic silver).

In the revised version of the manuscript we have added: "An external field B_{ext} of 3 mT, as is generated by Helmholtz coils, ..."

Referee: The caption of Fig. ED1 contains the non-existing word "minim".

Answer: Corrected.

Referee: In Eq. M8, $d_{\{x^2-y^2\}}$, brackets are missing around $(k_x^2 - k_y^2)$.

Answer: Corrected.

Response to Referee #2

We are grateful to the Referee for valuable comments and recommending our paper for publication.

Referee: The authors do not mention any x-ray characterisation of their La-doped crystals. Was this done? How much do the lattice parameters vary?

Answer: The values of the lattice constants are added to the sample description part: "The x-ray diffraction experiments performed on small powdered pieces cut from of each particular rod gave $a = 0.3867 \text{ nm}$, $c = 1.273 \text{ nm}$ for pure Sr₂RuO₄ and $a = 0.3865 \text{ nm}$, $c =$

1.274\$~nm for La-substituted sample."

46

Referee: The main result seems quite clear but the sample-to-sample variation in B_{TRSB} , as deduced from $\Delta\lambda$ is rather concerning. I appreciate that the authors state that they do not have a good explanation for this, but it might be helpful for future studies if they could provide a table of measured values of this parameter from their and other previously published studies to demonstrate the effect and provide a pointer for future work to explain this. Of course the problem with the usual group theory arguments is they can only tell you whether an effect is possible or forbidden, not what its strength is it is allowed.

50

Answer: The corresponding table is added.